

# Comparative chloroplast genomics of three species of *Bulbophyllum* section *Cirrhopetalum* (Orchidaceae), with an emphasis on the description of a new species from Eastern Himalaya

Mengkai Li[1,2], Lu Tang[3], Jianping Deng[1], Hanqing Tang[4], Shicheng Shao[5], Zhen Xing[2] and Yan Luo[1]

[1] Southeast Asia Biodiversity Research Institute, Chinese Academy of Sciences & Center for Integrative Conservation, Xishuangbanna Tropical Botanical Garden, Chinese Academy of Sciences, Mengla, Yunnan, China

[2] Resources & Environment College, Tibet Agriculture & Animal Husbandry University, Nyingchi, China

[3] Center for Gardening and Horticulture, Xishuangbanna Tropical Botanical Garden, Chinese Academy of Sciences, Mengla, Yunnan, China

[4] School of Resources and Environmental Science, Hubei University, Wuhan, China

[5] CAS Key Laboratory of Tropical Forest Ecology, Xishuangbanna Tropical Botanical Garden, Chinese Academy of Sciences, Mengla, Yunnan, China

Corresponding authors
Zhen Xing, xztibetan@163.com
Yan Luo, luoyan@xtbg.org.cn

## ABSTRACT

**Background**. Chloroplast (cp) genomes are useful and informative molecular markers used for species determination and phylogenetic analysis. *Bulbophyllum* is one of the most taxonomically complex taxa in Orchidaceae. However, the genome characteristics of *Bulbophyllum* are poorly understood.

**Methods**. Based on comparative morphological and genomic analysis, a new species *Bulbophyllum pilopetalum* from eastern Himalaya belonging to section *Cirrhopetalum* is described and illustrated. This study used chloroplast genomic sequences and ribosomal DNA (nrDNA) analysis to distinguish the new *Bulbophyllum* species and determine its phylogenetic position. An additional phylogenetic analysis was conducted using 74 coding sequences from 15 complete chloroplast genomes from the genus *Bulbophyllum*, as well as nrDNA sequences and two chloroplast DNA sequences from 33 *Bulbophyllun* species.

**Results**. The new species is morphologically similar to *B. pingnanense*, *B. albociliatum*, and *B. brevipedunculatum* in vegetative and floral morphology, but it can be distinguished by its ovate-triangle dorsal sepal without a marginal ciliate. The chloroplast genome of the new *Bulbophyllum* species is 151,148 bp in length, and includes a pair of inverted repeats (IRs) of 25,833 bp, a large single-copy region (LSC) of 86,138 bp, and a small single-copy region (SSC) of 13,300 bp. The chloroplast genome includes 108 unique genes encoding 75 proteins, 30 tRNAs, and four rRNAs. Compared with the cp genomes of its two most closely-related species, *B. pingnanense* and *B. albociliatum*, this chloroplast genome exhibited great interspecific divergence and contained several Indels that were specific to the new species. The plastid tree showed that *B. pilopetalum* is most closely-related to *B. pingnanense*. The phylogenetic tree based on combined

nrDNA and chloroplast DNA sequences indicated that section *Cirrhopetalum* was monophyletic and *B. pilopetalum* was a member of this section.

**Discussion**. The taxonomic status of the new species is strongly supported by cp genome data. Our study highlights the importance of using the complete cp genome to identify species, elucidate the taxonomy, and reconstruct the phylogeny of plant groups with complicated taxonomic problems.

## INTRODUCTION

*Bulbophyllum* Thouars is one of the largest orchid genera with 2,000–2,200 species that are distributed all over the pantropical regions, especially in the Paleotropics (*Pridgeon et al., 2014*). *Bulbophyllum* species have rounded pseudobulbs bearing one or two leaves and small flowers that often have colored sepals that are larger than the petals. This genus represents one of the most diverse genera in the Orchidaceae family, exhibiting a wide variety of growth forms and floral characteristics (*Gamisch & Comes, 2019*). The genus of *Bulbophyllum* has a complicated taxonomic history. *Bulbophyllum* was initially established in 1822 (*Thouars, 1822*). Nearly 50 independent and closely related genera were described, *e.g.*, *Cirrhopetalum* Lindl., *Drymoda* Lindl., *Monomeria* Lindl., *Trias* Lindl., and *Sunipia* Lindl. (*Pridgeon et al., 2014*; *Seidenfaden, 1979*). All of these allied genera have been combined with and separated from *Bulbophyllum* over the years. Molecular phylogenetic research recently supported a broad definition of *Bulbophyllum* that included all related genera (*Chase et al., 2015*; *Gravendeel, Fischer & Vermeulen, 2014*).

The *Cirrhopetalum* alliance in the genus *Bulbophyllum* is derived from the genus *Cirrhopetalum*, which is characterized by its sub-umbellate raceme inflorescence and significantly elongated lateral sepals that are twisted inward near base and connate (*Garay, Hamer & Siegerist, 1994*; *Seidenfaden, 1973*). It is a species-rich group, with ca. 150-240 species, mainly distributed in Asia (*Seidenfaden, 1973*; *Vermeulen, 2014*). *Vermeulen (2014)* considered it to be monophyletic and divided into 10 sections: *Acrochaene* (Lindl.) J. J. Verm., *Biflorae* Gary, Hamer & Siegerist, *Blepharistes* J. J. Verm., *Brachyantha* Rchb.f., *Cirrhopetaloides* Garay, Hamer & Siegerist, *Cirrhopetalum* (Lindl.) Rchb. f., *Emarginatae* Garay, Hamer & Siegerist, *Eublepharon* J. J. Verm., *Recurvae* Garay, Hamer & Siegerist, and *Rhytionanthos* (Garay, Hamer & Siegerist) J. J. Verm. However, *Hu et al. (2020)* combined DNA and floral characters to reconstruct the phylogenetic relationship in the Asian *Cirrhopetalum* alliance and further indicated that the majority of traditional sectional classifications were shown to be highly artificial. In the latest revision of *Bulbophyllum* in Flora of China, sections *Umbellata* Bentham & J. D. Hooker, *Corymbosa* (Blume) Aver., *Cirrhopetalum*, ''Zhonghuazu'', and ''Suihuazu'', which include 46 species, are involved in the *Cirrhopetalum* alliance (*Chen & Vermeulen, 2009*).

New species and records in the *Cirrhopetalum* alliance from China have been discovered more recently, *e.g.*, *B. picturatum* (Loddiges) H. G. Reichenbach, *B. pingnanense* J. F. Liu, S.

R. Lan & Y. C. Liang (*Liu et al., 2016*), *B. yunxiaoense* M. H. Li, J. F. Liu & S. P. Chen (*Li et al., 2017*), *B. yongtaiense* J. F. Liu, S. R. Lan & Y. C. Liang (*Liu et al., 2018*), *B. reflexipetalum* J. D. Ya, Y. J. Guo & C. Liu (*Ya et al., 2019*), *B. wendlandianum* (Kraenzl.) Dammer (*Jin, Li & Ye, 2019*). However, taxonomists still face difficulties in identifying the *Cirrhopetalum* alliance at the species level, and taxonomic uncertainties at the sectional level are majorly problematic and require revision.

During botanical explorations in the eastern Himalayas, we discovered a distinct species of *Bulbophyllum* from southeastern Xizang. Its sub-umbellate inflorescence and lateral sepals that are longer than its dorsal sepals strongly suggested that it should be assigned to the *Cirrhopetalum* alliance. Detailed examination of its morphology by living individuals and a survey of the literature (*Hsu & Chung, 2008*; *Chen & Vermeulen, 2009*; *Liu et al., 2016*) showed that it was morphologically related to *B. pingnanense*, *B. albociliatum* (Tang S. Liu & H. Y. Su) Seidenf., and *B. brevipedunculatum* T. C. Hsu & S. W. Chung, but it is distinguished from them by its smooth dorsal sepals.

Many new species have been described in recent years, and molecular markers have greatly enhanced our understanding of species delimitation (*Li et al., 2017*; *Liu et al., 2018*). Chloroplast (cp) genome sequences have contributed to solving phylogenetic and taxonomic problems in many plant groups and provided barcodes for identifying species (*Niu et al., 2018*; *Pfanzelt, Albach & von Hagen, 2019*). It is possible to obtain a complete cp genome for plants using next-generation sequencing techniques (*Wilkinson et al., 2017*). However, there is a lack of genetic and molecular data for this diverse genus of *Bulbophyllum*. The complete cp genome may be used to examine sequence divergence that could help resolve taxonomic uncertainties in the *Bulbophyllum* genus.

In this study, we sequenced and performed a comparative analysis of the complete cp genome of the new species and its closely related species, *B. pingnanense* and *B. albociliatum*. Furthermore, molecular phylogenetic analysis using the complete cp genome and nrDNA internal transcribed spacer (ITS) was conducted to determine the phylogenetic position of this new species. This study aims to confirm that our newly collected specimen is an undescribed species based on cp genome sequence divergence and use whole cp genome data to resolve taxonomic uncertainties within the *Cirrhopetalum* alliance.

## MATERIALS & METHODS

### Morphological analyses

Morphological characteristics of the new species were studied based on the study of living materials. Images of flowering plants were taken in the field. In addition, the morphological study included a comparison of the new species with other species, based on the Flora of China (*Chen & Vermeulen, 2009*) and a bibliography of related species (*Hsu & Chung, 2008*; *Liu et al., 2016*). A total of 13 diagnostic characteristics of the new species were compared to those of three closely related species. Voucher Specimens were deposited in the Herbarium of Xishuangbanna Tropical Botanical Garden (HITBC), Chinese Academy of Sciences, while the living individuals were preserved in the nursery.

### New botanical taxonomic name

The electronic version of this article in Portable Document Format (PDF) will represent a published work according to the International Code of Nomenclature for plants (ICN), and hence the new names contained in the electronic version are effectively published under that Code from the electronic edition alone. In addition, new names contained in this work which have been issued with identifiers by IPNI will eventually be made available to the Global Names Index. The IPNI LSIDs can be resolved and the associated information viewed through any standard web browser by appending the LSID contained in this publication to the prefix "http://ipni.org/". The online version of this work is archived and available from the following digital repositories: PeerJ, PubMed Central SCIE, and CLOCKSS.

### Sampling, DNA extraction and sequencing

Fresh leaves of *B. pilopetalum* (*Voucher: Luo et al. 3521*) and *B. albociliatum* (*Voucher: Xu & Wang 202106*) were collected from the living collections at the nursery of Xishuangbanna Tropical Botanical Garden, Chinese Academy of Sciences and Yunnan Fengchunfang Biotechnology Co. Ltd. Additionally, we downloaded the available complete cp genomes of 12 *Bulbophyllum* species and DNA sequences (cpDNA: *mat* K and *psb* A-*trn* H; nrDNA: ITS) of 30 species of the *Cirrhopetalum* alliance of *Bulbophyllum* from GenBank (Table S1) to analyze genomic characteristics and phylogenetic relationships. The methods of both DNA extraction and sequencing were followed *Tang et al. (2021)*.

### Chloroplast genome assembly and annotation

GetOrganelle1.7.5 was used for de novo cp genome and the nrDNA sequence (18S-ITS1−5.8S-ITS2-26S) assembly with default parameters (*Jin et al., 2020*). CPGAVAS2 (*Shi et al., 2019*) and GeSeq (*Tillich et al., 2017*) were used to annotate the assembled cp genome using default parameters to predict protein-coding, rRNA, and tRNA genes. The sequence identity and annotation results were checked and the manual corrections and codon positions were adjusted by comparing them with *Bulbophyllum* species present in the database using GeneiousPrime 2021. The length of the whole plastome, number of genes, categories of genes, and GC content were analyzed in GeneiousPrime. The genome map of the species was illustrated with the help of OGDRAW (*Greiner, Lehwark & Bock, 2019*). The annotated cp genome and nrDNA sequences were submitted to GenBank (Table S1).

### Comparative genomic and phylogenetic analysis

Comparative genomic analysis, including cp genomes sequence divergence, mutations and indels, the borders, and measurement of the nucleotide diversity value (Pi), was conducted as previously described in *Tang et al. (2021)*. The nucleotide identity of the cp genomes of all three *Bulbophyllum* species was calculated using GeneiousPrime.

We selected the complete cp genomes of the two newly sequenced *Bulbophyllum* species, 13 *Bulbophyllum* species, and two species from *Dendrobium* (Table S1) downloaded from GenBank to reconstruct the phylogenetic tree. Phylogenetic analysis was performed by 74 protein-coding sequences of the cp genome, using Maximum Parsimony (MP), Maximum

Likelihood (ML), Bayesian (BI) analyses. The methods of constructing phylogenetic tree were previously described in *Tang et al. (2021)*. Additionally, to estimate the systematic position of the new species within the *Cirrhopetalun* alliance, we used nrDNA ITS and two cp DNA sequences (*matK*, *psb* A-*trnH*) from 30 species of the *Cirrhopetalum* alliance available from GenBank to reconstruct their phylogenetic relationships (Table S1). *Dendrobium* species were used as the outgroups. The phylogenetic relationship analysis was performed using ML methods combined with nr DNA and cp DNA sequences.

# RESULTS

## Taxonomy

*Bulbophyllum pilopetalum* M. K. Li, J. P. Deng & Y. Luo sp. nov. (Figs. 1–2)

**Etymology.** The specific epithet "*pilopetalum*" refers to the petal margins dense with white ciliate.

**Holotype.** CHINA. XIZANG: Bomi County, Tongmai Village, on the trunks of trees or sometimes on rocks in the humid evergreen broad-leaved forest, 2100 m, Mar. 10, 2021, *Y. Luo, M. K. Li & L. Tang* 3521, fl. [holotype (HITBC0074007) & isotype (HITBC0074008), HITBC).

**Diagnosis.** Similar to *B. brevipedunculatum*, *B. albociliatum*, and *B. pinganense*, but differs by having a shorter peduncle, triangle dorsal sepal with margin entire, oblong petals, and divergent lateral sepals twisted slightly near the apex.

**Description.** Short-creeping epiphyte. Rhizome ca. 0.7 mm in diam. Pseudobulbs 1.2–2 cm apart on rhizome, subglobose or ovoid, 5–7 mm, 3–5 mm in diam., young enveloped by brownish scarious sheaths, old with fibrous remnants at the base, with a terminal leaf. Leaf sessile; blade oblong to ovate-lanceolate, 1– 4× 0.7–0.8 cm, rigid, base cuneate, apex obtuse. Scape from the base of pseudobulb, 3–5 mm, slightly shorter than pseudobulb, sub-umbel 1- or 2-flowered; peduncle less than 5 mm, enclosed in 2 or 3 membranous sheaths; floral bracts ovate-lanceolate, ca. 3 mm, apex shortly acute. Pedicel and ovary ca. 5 mm. Flowers orangeish, with orange sepals and red petals. Sepals free, thickly textured; dorsal sepal concave, ovate-triangular, 3.5–4 mm × ca. 2 mm, margin entire, apex acute and beaked; lateral sepals oblong, 7–8 mm × 2–3 mm, twisted inward near the base, free and slightly divergent, margin entire, apex obtuse. Petals oblong, 1.5–2 mm × ca. one mm, margin densely long white ciliate, apex rounded; lip orange, recurved, oblong-lanceolate, ca. 2 × 1.1 mm, basal half grooved, base attached to the end of column foot by a moveable joint, apex obtuse; disk nearly smooth adaxially. Column pale white, ca. 1.5 mm, stout, with a distinct foot, conspicuously winged; stelidia filiform, acute, ca. 0.5 mm; foot curved, ca. 1.5 mm, with free part ca. 0.8 mm; anther cap subglobose, orange, fimbriate at apex; pollinia 2, ovoid, each with one deep groove dividing pollinium into two unequal particles, without appendage. Ovary 5–6 ribbed, glabrous, ca. 2 mm long, 0.8 mm in diam.

**Conservation status.** *Bulbophyllum pilopetalum* is known only from the type locality, and only two small populations of ca. 10 individual plants within a narrow area were discovered. The appropriate data on the abundance and distribution of this species is lacking. It can be estimated IUCN Red List status—Data Deficient (DD) (IUCN 2012).

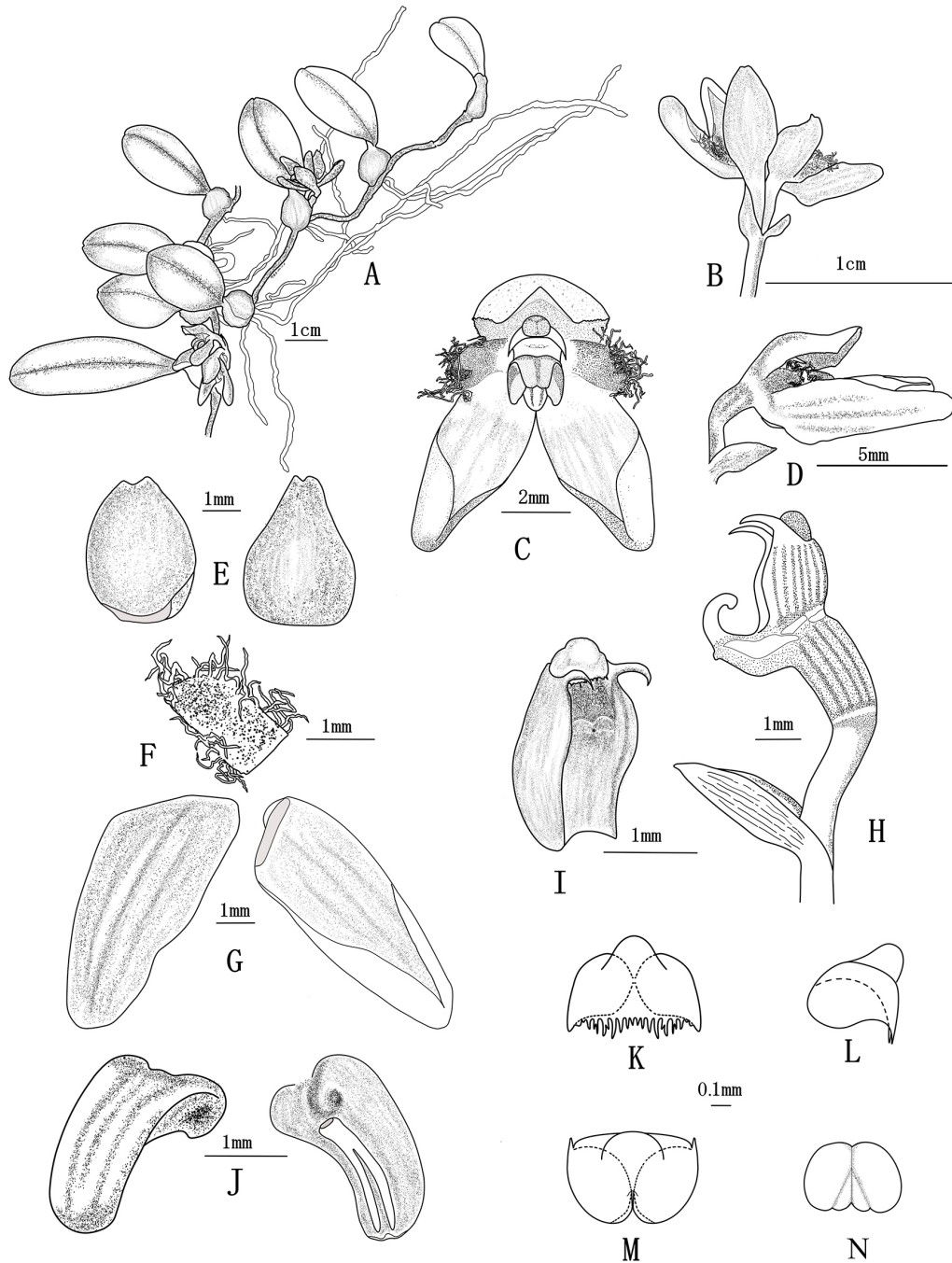

**Figure 1** ***Bulbophyllum pilopetalum*** M. K. Li, J. P. Deng & Y. Luo. (All drawn from Y. Luo et al. 3521 by J. P. Deng). (A) Flowering plant; (B) inflorescence; (C) flower, frontal view; (D) flower, side view; (E) dorsal sepal, adaxial and abaxial top view; (F) petal, adaxial top view; (G) lateral sepal, adaxial and abaxial top view; (H) Column and ovary, side view; (I) column, bottom view; (J) lip, side view and bottom view; (K–M) anther cap, frontal view, top view and bottom view; (N) pollinia, frontal view.

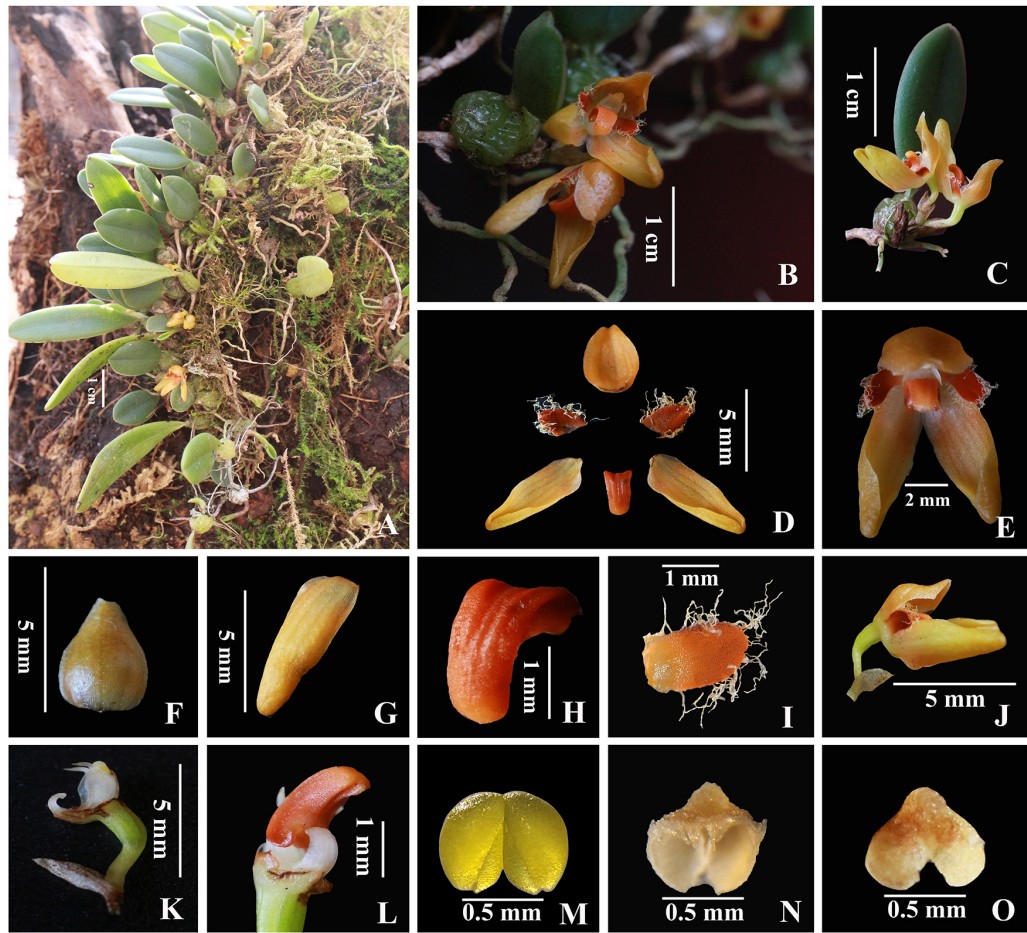

**Figure 2** *Bulbophyllum pilopetalum.* M. K. Li, J. P. Deng & Y. Luo. (Photos from Y. Luo et al. 3521 by J. P. Deng in Bomi, Tibet, China). (A) Habit; (B–C) plant and inflorescence; (D) sepals, petals and lip frontal view; (E) flower, frontal view; (F) dorsal sepal, abaxial top view; (G) lateral sepal, abaxial top view; (H) lip, side view; (I) petal, top view; (J) flower and floral bract, side view; (K) column and ovary, side view; (L) lip and Column foot, bottom view; (M) pollinia, frontal view; (N-O) anther cap, top view and bottom view.

**Notes.** Morphologically, *B. pilopetalum* is allied to some taxa from the eastern coast of China, namely *B. pingnanense*, *B. albociliatum,* and *B. brevipedunculatum* by its dwarf plants, pseudobulbs separated by rhizomes about 1 to 2 cm, orangeish flowers and petals with ciliate margin, elongated lateral sepals twisted inward near the base. These three species are placed under sect. *Cirrhopetalum* (*Chen & Vermeulen, 2009*; *Liu et al., 2016*). Our plant has similar characteristics and should also belong to this section. However, it differs from other species by having an ovate-triangular dorsal sepal with a conspicuous beak and entire margin, and oblong petals. The differences between these four related species are shown in Table 1. *B. pilopetalum* may be most closely related to *B. brevipedunculatum*, which has a short scape and 1–3 flowers, but our plant can be distinguished by its oblong lateral sepals,

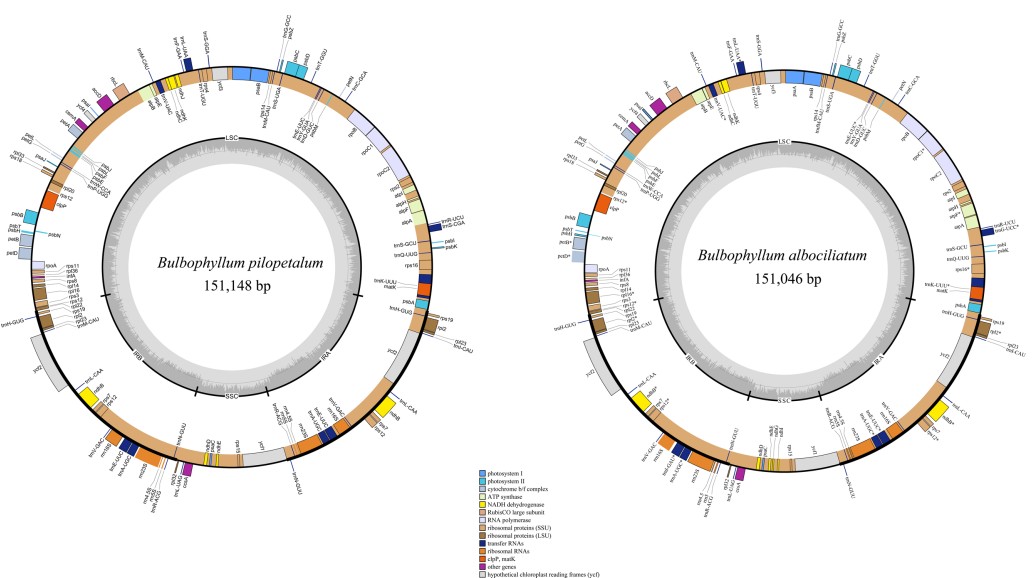

**Figure 3** **Chloroplast genome map for *Bulbophyllum pilopetalum* M. K. Li, J. P. Deng & Y. Luo (A) and *B. albociliatum* (T. S. Liu et H. J. Su) Seidenf. (B).** Genes located outside the outer rim are transcribed in a counterclockwise direction, whereas genes inside the outer rim are transcribed in a clockwise direction. The colored bars indicate known different functional groups. The dashed gray area in the inner circle shows the percentage GC contents of the corresponding genes. LSC, SSC, and IR denote large single copy, small single copy, and inverted repeat, respectively.

open divergent. *B. pilopetalum* differs from *B. pingnanense* and *B. albociliatum* by having a shorter scape, lateral sepals, and fewer flowers.

## Characteristics of *Bulbophyllum* chloroplast genome

The lengths of the cp genomes of *B. pilopetalum* and its two closely related species, *B. albociliatum* and *B. pingnanense* (MW822749), are 151,148 bp, 151,224 bp, and 151,046 bp, respectively (Fig. 3, Table 2). Three cp genomes displayed a typical quadripartite structure and contained a pair of inverted repeats (IRs, 25,833 to 25,855 bp), separated by a large single copy region (LSC, 85,956 to 86,138 bp) and a small single copy region (SSC, 13,300 to 13,441 bp), respectively (Table 2). The overall GC contents of *B. pilopetalum* was 36.9% and that of both *B. pingnanense* and *B. albociliatum* was 37%. Three of the cp genomes each consistently contained 109 unique genes which were arranged in the same order across the genomes, including 75 protein-coding (CDS), 30 tRNA, and four rRNA genes (Table 3). The gene map was shown in Fig. 3. Fourteen genes (eight CDS and six tRNA) contained one intron in three *Bulbophyllum* species and three genes (*ycf3*, *clpP*, and *rps12*) contained two introns (Table 3).

## Comparative genomic analysis
### Sequence divergence

The comparative sequence analysis of three *Bulbophyllum* species revealed that the cp sequences were highly conserved across the three taxa with a few variable regions (Fig. 4).

**Table 1** Morphological comparison of *Bulbophyllum pilopetalum* and its closely related species.

| Characters | B. pilopetalum | B. brevipedunculatum[a] | B. albociliatum[b] | B. pingnanense[c] |
|---|---|---|---|---|
| Pseudobulbs apart | 1.2–2 cm | 1–2 cm | 2 cm | 0.6–2.5 cm |
| Pseudobulb | subglobose or ovoid, 5–7 mm, 3–5 mm in diam. | elongate ovoid, 7–10 mm, 3–5 mm in diam. | elongate-ovoid, 10–13 mm, 5–7 mm in diam. | obvate-elliptic, 0.5–1.7 cm, 3–6 mm in diam. |
| Leaf | oblong to ovate-lanceolate, 1–4 × 0.7–0.8 cm, apex obtuse, emarginate | oblong to linear-oblong, 1–7 cm × 0.7–0.8 cm, apex obtuse to retuse | oblong, oblanceolate or obovate, 2–4 cm × 0.7–1 cm, apex obtuse or rounded | oblong to linear-oblong, 1.8–6.6 × 0.6–1.2 cm, apex obtuse to retuse |
| Scape | scape slightly shorter than pseudobulb, 3–5 mm, 1–2 flower | scape shorter than pseudobulb, 5–7 mm, 1- 3 flowered | scape much longer than pseudobulb, ca. 4–6 cm, 5–6 flower | scape much longer than pseudobulb, ca. 1.1 cm, 3–5 flower |
| Floral bract | ovate-lanceolate, ca. 3 mm | elongate triangular, ca. 3 mm | narrowly triangular, ca. 3 mm | triangular, ca.2–3 mm |
| Pedicel and ovary | 5 mm | 4–5 mm | 5 mm | 4 mm |
| Flower | orangeish, with red petals | reddish | rubescent, with reddish yellow lateral sepals | orange red |
| Dorsal sepal | concave, ovate-triangular, 3.5–4 × ca. 2 mm, entire, apex acute and beaked | strongly concave, elliptic, ca. 3.5 mm × 2 mm, apex acuminate, short white ciliate | rubescent, strongly concave, elliptic, 3–4 mm × 2 mm, apex rounded, long white ciliate | concave, ovate, abaxially papillose, 5 × 3 mm, apex obtuse or acute margins long white ciliate, |
| Lateral sepal | oblong, 7–8 × 2–3 mm, apex obtuse, open divergent | near rectangular, 5–7 mm × 2–3 mm, apex mucronate, often connate | lanceolate, 7–9 mm × 2 mm, incurved along margins, often connate | laceolate, 10–12 × 2 mm, apex acute, open divergent or connate |
| Petal | oblong, ca. 1.5–2 mm × 1 mm, margins densely with white ciliate, apex rounded | elliptic, ca. 2 × 1.2 mm, white ciliate, apex rounded, | elliptic, ca. 2 mm × 1.2 mm, apex rounded, long white ciliate | ovate, ca. 2.7–3.0 × 1.2–2.0 mm, margins long white ciliate, apex round |
| Lip | orange, oblong-lanceolate, ca, 2 × 1.1 mm, apex rounded obtuse, | dark red, horn-like, ca. 2 mm, apex rounded obtuse | rubescent, horn-like, ca. 1.5 mm, recurved, adaxial surface of lip partly papillose | ovate-triangular, ca. 3 mm, abaxially deeply grooved |
| Column | pale white, 1.5 mm | pale yellow, ca. 1.5 mm, | yellow, ca. 1.5 mm | yellow, subterete, ca. 1–2 mm |
| Anther cap | subglobose | cordate | cordate | subglobose |

Notes.
[a](*Hsu & Chung, 2008*).
[b](*Chen & Vermeulen, 2009*).
[c](*Liu et al., 2016*).

**Table 2 The basic characteristics of the chloroplast genomes of three *Bulbophyllum* species.**

|  | *B. pilopetalum* | *B. pingnanense* | *B. albociliatum* |
|---|---|---|---|
| genome size (bp) | 151,148 | 151,224 | 151,046 |
| LSC length (bp) | 86,138 | 86,017 | 85,956 |
| SSC length (bp) | 13,300 | 13,441 | 13,384 |
| IR length (bp) | 25,833 | 25,855 | 25,853 |
| Number of genes | 109 | 109 | 109 |
| Protein-coding genes | 75 | 75 | 75 |
| tRNA genes | 30 | 30 | 30 |
| rRNA genes | 4 | 4 | 4 |
| GC content (%) | 36.9 | 37 | 37 |
| GC content in LSC (%) | 34.4 | 34.4 | 34.5 |
| GC content in SSC (%) | 29.1 | 29.2 | 29.2 |
| GC content in IR (%) | 43.2 | 43.2 | 43.2 |
| GC content in Protein-coding (%) | 37.9 | 37.9 | 37.8 |
| GC content in tRNA (%) | 52.8 | 53 | 53 |
| GC content in rRNA IR (%) | 54.9 | 54.9 | 54.9 |

The sequences were more conserved in IR regions, whereas, most of the divergence detected were found in LSC and SSC regions (Fig. 4).

### Mutations and indels

The number of mutations and indel events was compared between the cp genomes of *B. pilopetalum* and its two closely related species, respectively. There were 1,952 mutations and 445 indels calculated between *B. pilopetalum* and *B. albociliatum*, with the highest indel rate of 0.33 indels per 100 bp in the intergenic regions, while the highest mutation rate was 1.70 mutations per 100 bp in introns (Table 4). There were 1,575 mutations and 239 indels calculated between *B. pilopetalum* and *B. pingnanense*, with the highest indel rate of 0.22 indels per 100 bp in introns, while the highest mutation rate was 1.25 mutations per 100 bp in the intergenic regions (Table 4).

We further analyzed the details of indels (indel size>10 bp) of the three *Bulbophyllum* species based on multiple whole cp genome alignments. DNA indels were found in 36 positions in at least one of the cp genomes, predominately in the intergenic regions, as well as in introns or in the coding regions. *B. pilopetalum* and *B. pingnanense* shared 11 indels, and *B. pilopetalum* and *B. albociliatum* shared 11 indels as well. There were 14 unique indels in *B. pilopetalum*, including *ycf1* (111 bp, coding region), *psbI-atpA* (50 bp), *ycf3* (29 bp, intron), *psbZ-rps14* (24 bp), *rpl12-rpl32* (20 bp), *ycf1-rpl2* (20 bp), and *psbM-petD* (12 bp) (Table S2).

### Border contraction and extension

We analyzed the IR/single copy (SC) region border positions and their adjacent genes in the three *Bulbophyllum* cp genomes (Fig. 5). In all cp genomes of the three *Bulbophyllum* species, the genes *rpl22*, *rps19*, *trnN* -GUU, *rpl32*, *ycf1*, and *psbA* were located at the junction of the LSC/IRb, IRb/SSC, SSC/IRa, and IRa/LSC borders (Fig. 5). The border regions of

**Table 3** Gene contents in the chloroplast genomes of *Bulbophyllum pilopetalum*, *B. pingnanense* and *B. albociliatum*.

| Group of genes | Gene names |
|---|---|
| 1 Photosystem I | psaA, psaB, psaC, psaI, psaJ |
| 2 Photosystem II | psbA, psbB, psbC, psbD, psbE, psbF, psbH, psbI, psbJ, psbK, psbL, psbM, psbN, psbT, psbZ |
| 3 Cytochrome b/f complex | petA, petB*, petD*, petG, petL, petN |
| 4 ATP synthase | atpA, atpB, atpE, atpF*, atpH, atpI |
| 5 NADH dehydrogenase | ndhA*, ndhB*(×2), ndhC, ndhD, ndhE, ndhF, ndhG, ndhH, ndhI, ndhJ, ndhK |
| 6 RubisCO large subunit | rbcL |
| 7 RNA polymerase | rpoA, rpoB, rpoC1*, rpoC2 |
| 8 Ribosomal proteins(SSU) | rps2, rps3, rps4, rps7(×2), rps8, rps11, rps12**(×2), rps14, rps15, rps16*, rps18, rps19(×2) |
| 9 Ribosomal proteins(LSU) | rpl2(×2), rpl14, rpl16*, rpl20, rpl22, rpl23(×2), rpl32, rpl33, rpl36 |
| 10 Other genes | clpP**, matK, accD, ccsA, infA, cemA |
| 11 Proteins of unknown function | ycf1, ycf2(×2), ycf3**, ycf4 |
| 12 Ribosomal RNAs | rrn4.5S(×2), rrn5S(×2), rrn16S(×2), rrn23S(×2) |
| 13 Transfer RNAs | trnA-UGC*(×2), trnC-GCA, trnD-GUC, trnE-UUC, trnF-GAA, trnfM-CAU, trnG-GCC, trnG-UCC*, trnH-GUG(×2), trnI-CAU(×2), trnI-GAU*(×2), trnK-UUU*, trnL-CAA(×2), trnL-UAA*, trnL-UAG, trnM-CAU(×2), trnN-GUU(×2), trnP-UGG, trnQ-UUG, trnR-ACG(×2), trnR-UCU, trnS-GCU, trnS-GGA, trnS-UGA, trnT-GGU, trnT-UGU, trnV-GAC(×2), trnV-UAC*, trnW-CCA, trnY-GUA |

**Notes.**
*Gene containing one intron.
**Gene containing two introns, (x2) Gene has two copies.

**Table 4** Mutations, indel events, and mutation rates comparison of the chloroplast genomes of *Bulbophyllum pilopetalum* and its closely related species.

| | *B. pilopetalum* vs *B. albociliatum* | | | *B. pilopetalum* vs *B. pingnanense* | | |
|---|---|---|---|---|---|---|
| | **CDS** | **IGS** | **intron** | **CDS** | **IGS** | **intron** |
| Indels | 125 | 262 | 58 | 91 | 95 | 53 |
| Mutations | 438 | 1142 | 372 | 477 | 993 | 105 |
| Total length | 72,632 | 79,255 | 21,939 | 72,209 | 79,235 | 23,828 |
| indels/100 bp | 0.17 | 0.33 | 0.26 | 0.13 | 0.12 | 0.22 |
| mutations/100 bp | 0.60 | 1.44 | 1.70 | 0.66 | 1.25 | 0.44 |

**Notes.**
CDS, protein-coding genes; IGS, intergenic spacers.

cp genomes were similar across the three *Bulbophyllum* species. There was 37 to 43 bp extension of *rpl22* gene into the IRb region. There was 393–422 bp between *trnN*- GUU and the LSC/IRa border, while the *rpl32* generated a distance of 89–115 bp to another LSC/IRa junction. The *ycf1* gene crossed the SSC/IRa junction expanding 61–91 bp to IRa

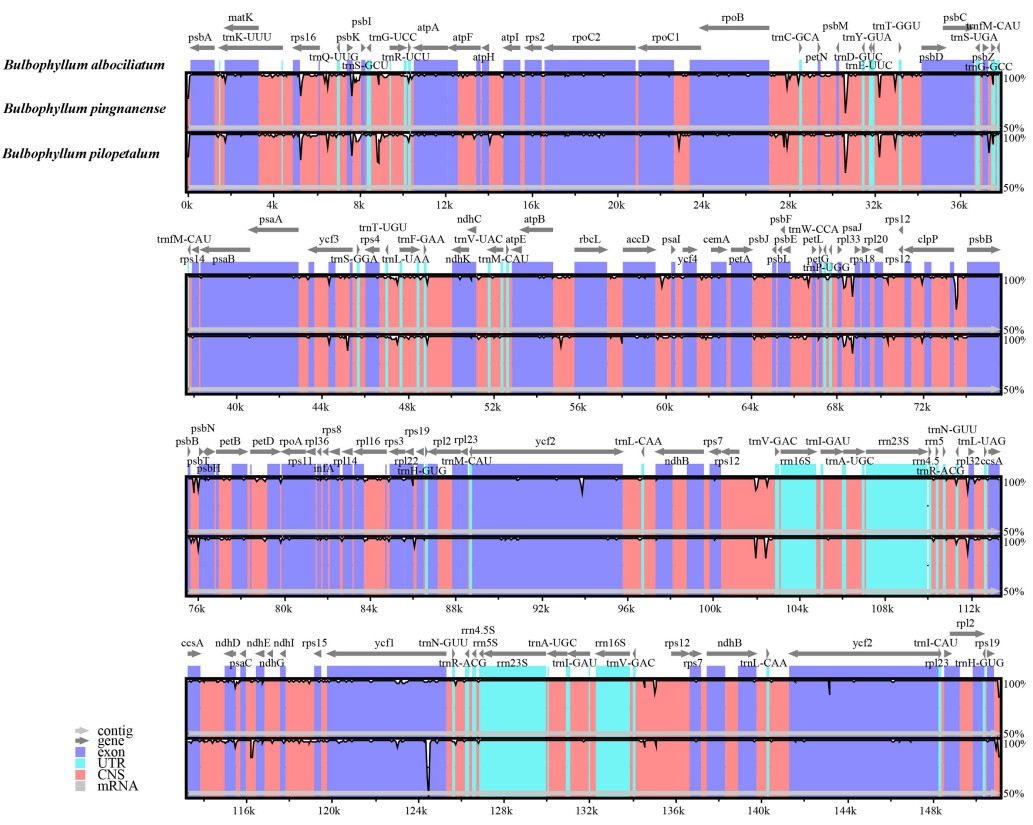

**Figure 4** Alignment of the cp genome sequences of three *Bulbophyllum* species, with *B. albociliatum* as a reference. The *X*-axis corresponds to coordinates within the cp genome. The *Y*-axis shows the percentage identity in the range 50% to 100%.

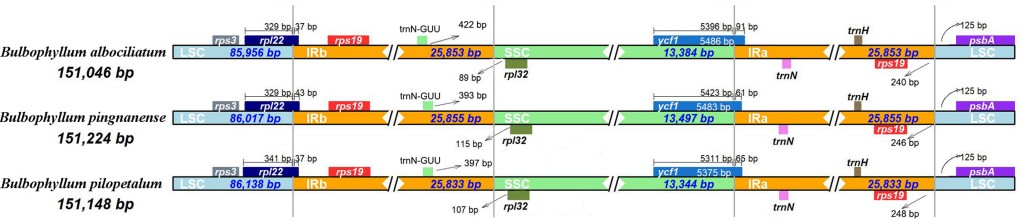

**Figure 5** Comparison of the borders between neighboring genes and junctions of LSC, SSC, and IR regions of the chloroplast genomes in three *Bulbophyllum* species. Boxes above or below the main line indicate genes adjacent to borders.

regions in three species. There was 240–248 bp between *rps19* and the LSC/IRa border, and the *psbA* generated a distance of 125 bp to another LSC/IRa junction.

## Nucleotide diversity and identity

The nucleotide diversity value (Pi) of the cp genome was calculated using DnaSP between the new species and its closely related species to understand their sequence divergence. The

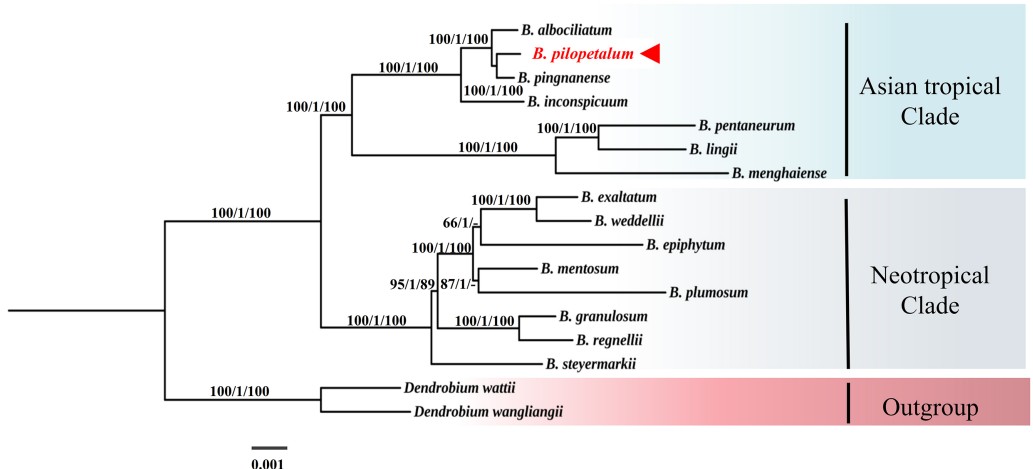

**Figure 6** **Phylogenetic position of *Bulbophyllum pilopetalum* based on maximum likelihood (ML), Bayesian inference (BI) and maximum parsimonious (MP) analysis of 74 protein-coding chloroplast genes from *Bulbophyllum* species.** The numbers above branches represent bootstrap percentage (BP) of ML/posterior probability (PP) of BI /BP of MP ("-" indicates a BP < 90%).

Pi values between *B. pilopetalum* and *B. albociliatum* ranged from 0 to 0.0179, which were higher compared to those between *B. pilopetalum* and *B. pingnanense,* with the Pi values ranging from 0 to 0.0150 (Fig. S1). The nucleotide identity of the cp genome was 99.124% between *B. pilopetalum* and *B. albociliatum*, and 99.229% between *B. pilopetalum* and *B. pingnanense.*

## Phylogenetic position of new species

In the phylogenetic tree of 74 protein-coding genes of the cp genome using ML, MP, and BI analyses (Fig. 6), the *Bulbophyllum* genus was monophyletic and formed two clades, the Asian tropical and Neotropical *Bulbophyllum*. The Asian tropical clade formed two subclades: three species from *B.* sect. *Macrocaulia* (Blume) Aver., and *B. inconspicuum* Maxim. (Native to Japan) was a sister to these three closely related species, which strongly supported that these three species were monophyletic. Within this clade, the sister relationship between *B. pilopetalum* and *B. pingnanense* was highly supported.

ML analyses were conducted based on nrDNA ITS and two cp DNA sequences from 33 species of the *Cirrhopetalum* alliance (Fig. 7). Phylogenetic analysis showed seven clades were grouped, but they were not well consistent with sectional classification by *Chen & Vermeulen (2009)* and *Vermeulen (2014)*. The clade included *B. dayanum* from sect. *Acrochaene* formed the base of the *Cirrhopetalum* alliance. The ML trees showed a strongly supported clade (CIR) grouping 10 species from sect. *Cirrhopetalum* together. Within this clade, *B. pilopetalum* and *B. pingnanense* were placed together as sisters. Six species from sect. *Brachyantha* were divided into two subclades. Four other clades were recognized with strong support, but each included species from two or three sections (Fig. 7).

## DISCUSSION

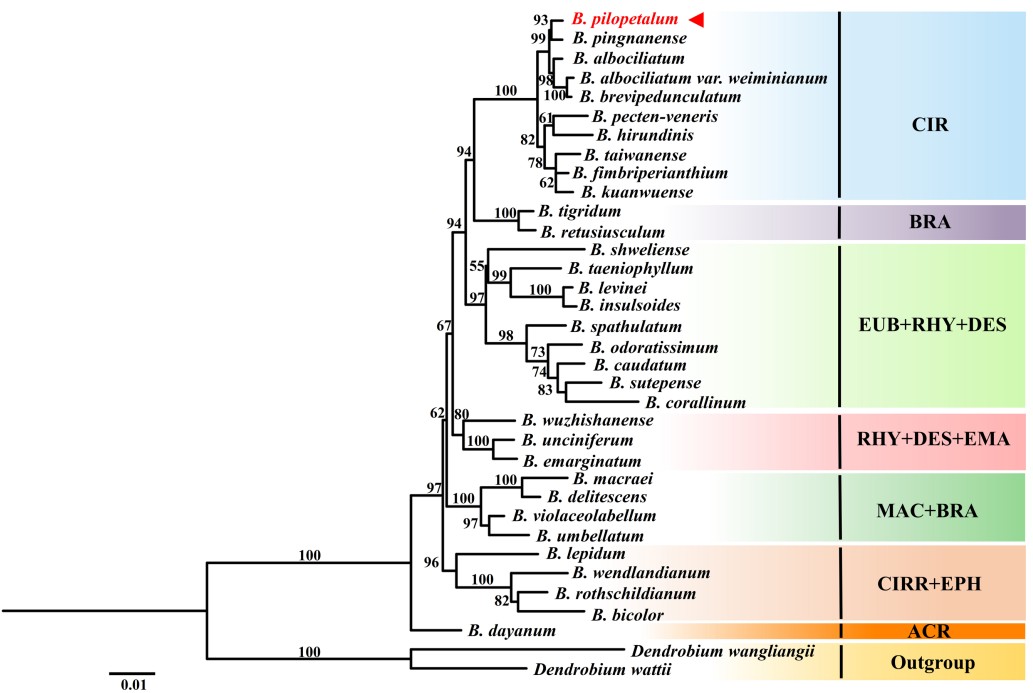

**Figure 7** Phylogenetic position of *Bulbophyllum pilopetalum* based on Maximum Likelihood analysis of the combined nuclear (ITS) and chloroplast DNA sequence *(matK* and *psbA-trnH*). The numbers above branches are ML bootstrap support values (BP); sectional classification of taxa (according to *Chen & Vermeulen, 2009*; *Vermeulen, 2014*) is indicated by three-letter abbreviations following the taxon name: ACR: sect. *Acrochaene*, BRA: sect. *Brachyantha*, CIR: sect. *Cirrhopetalum*, CIRR: sect. *Cirrhopetaloides*, DES: sect. *Desmosanthes*, EPH: sect. *Ephippium*, EUB: sect. *Eublepharon*, EMA: sect. *Emarginatae*, MAC: sect. *Macrostylida*, RHY: sect. *Rhytionanthos*.

## Evidence for species delimitation based on chloroplast genomes

The cp genome sequence provided comprehensive and adequate information to better understand the phylogeny and improve species identification efficiency (*Niu et al., 2018*; *Pfanzelt, Albach & von Hagen, 2019*). Comparative cp genomic analysis of three species from *B*. sect. *Macrocaulia* showed cp genomes of sect. *Macrocaulia* species had similar structure and gene content and shared a number of indels, which mainly contributed to its monophyly (*Tang et al., 2021*). Cp genomes of eight Neotropical *Bulbophyllum* species provide good resolution for the sectional classification of Neotropical *Bulbophyllum* (*Zavala-Páez et al., 2020*). The comparison of the cp genome structure between *B. pilopetalum*, *B. albociliatum,* and *B. pingnanense* revealed that the gene composition, gene structure, and the number of genes were found to be similar, and the differences between the border contraction and extension were not significant. The nucleotide identity of the cp genome between the three species ranged from 99.124–99.270%, with a slightly higher similarity of 99.229% between *B. pilopetalum* and *B. pingnanense*. The cp genome showed high sequence divergence between *B. pilopetalum* and its two closely related species: Pi value of 0−0.0179 between *B. pilopetalum* and *B. albociliatum* and 0−0.0150 between *B. pilopetalum* and *B. pingnanense*. The mutations and indels were frequently found between

the cp genomes of three species, whereas 14 unique indels (indel size>10 bp) were found in *B. pilopetalum*. The differences in cp genome structure distinguish *B. pilopetalum* from other closely related species, further supporting its status as a distinct species. The indels specific to this species can be used as diagnostic DNA sequence characteristics (Table S2).

## The new species could be placed in *B. sect. Cirrhopetalum*

Section *Cirrhopetalum* belongs to the core *Cirrhopetalum* alliance (*Hu et al., 2020*). However, its taxonomic status and circumscription remain controversial. Sect. *Cirrhopetalum* is characterized by sub-umbellate raceme, lateral sepals longer than the dorsal sepal, twisting and connected lateral sepals, and hairy dorsal sepal and petals (*Seidenfaden, 1979*). Several researchers have proposed different classifications of the sect. *Cirrhopetalum*. For example, sect. *Cirrhopetalum* was listed with 42 species in Flora of China (*Tsi, Chen & Luo, 1999*), but was later revised to 17 species (*Chen & Vermeulen, 2009*). The molecular phylogeny of *Hu et al. (2020)* showed that the previous sect. *Cirrhopetalum* was polyphyletic and divided into two major clades, B and D7 (*Hu et al., 2020*). To determine the phylogenetic position of the new species, we selected nrDNA ITS and cp DNA sequences from eight *Bulbophyllum* species from a highly supported subclade of sect. *Cirrhopetalum* (Subclade D7), regarded as reliable members of sect. *Cirrhopetalum* according to *Hu et al. (2020)*, and 23 other representative species from the *Cirrhopetalum* alliance, to reconstruct the phylogenetic tree. The phylogenetic analysis strongly supported that the new species is closely related to *B. pingnanense* within the sect. *Cirrhopetalum* clade (Fig. 7). The cp genome also supported that the new species is a sister to *B. pingnanense* (Fig. 6).

However, the new species is most closely and morphologically similar to *B. brevipedunculatum*. This is perhaps because morphological characteristics have not been fully understood between these species. In the phylogenetic tree based on nrDNA ITS and cp DNA sequences, sect. *Cirrhopetalum* was supported as a monophyletic clade, excluding *B. lepidum*, *B. wendlandianum,* and *B. rothschildianum* (Fig. 7). Within this clade, *B. albociliatum*, *B. brevipedunculatum,* and *B. albociliatum* var. *weiminianum* T.P. Lin & L. L. K. Huang formed a subclade, indicating that the three taxa were closely related. *B. brevipedunculatum* was treated as a variety of *B. albociliatum*: *B. albociliatum* var. *brevipedunculatum* (T. C. Hsu & S.W. Chung) W. M. Lin (*Lin & Wang, 2014*). The phylogenetic analysis in this study seems to support Lin's taxonomic treatment. However, other clades do not correspond with the taxa elucidated in Chen & Vermeulen's previous classification, suggesting that the current sectional classifications of the *Cirrhopetalum* alliance were only artificial and are in need of taxonomic revision.

The CIR clade in our study (Fig. 7) or the CIRRII (subclade D7) in the study of *Hu et al. (2020)* is well supported because a monophyletic group could be regarded as the circumscription of sect. *Cirrhopetalum*. Furthermore, although the sampling in the phylogenetic analysis of cp genome in this study was limited, the species *B. inconspicuum* distributed in Japan also clustered into a clade with our species. All these species have similar vegetative features, yellow, orange, or reddish flowers and dorsal sepals/petals with margins ciliate. These morphological features, such as dwarf epiphyte, short sub-umbellate inflorescence, yellowish flower, and ciliate sepals and petals, might be the diagnostic

characteristics for sect. *Cirrhopetalum*. Dorsal sepal/petal with ciliate margin was considered to be the ancestral characteristics in the *Cirrhopetalum* alliance. However, our species has dorsal sepals with entire margin, which is a gradual transition from smooth to ciliate (or ciliate to smooth) dorsal sepal that might occur in this species. Other members included in previous sect. *Cirrhopetalum*, determined by *Chen & Vermeulen (2009)* and *Vermeulen (2014)* (*e.g.*, species in the B clade of *Bulbophyllum* phylogeny of *Hu et al. (2020)*, which was previously placed in sect. *Cirrhopetalum*), should be considered to revise their taxonomic status. In order to further revise sect. *Cirrhopetalum*, the *Cirrhopetalum* alliance, even the *Bulbophyllum* genus, we need to integrate comprehensive morphological characteristics (*e.g.*, vegetative leaf and pseudobulbs, pollinia, and seeds) and more suitable molecular markers (*e.g.*, cp genomes).

## ACKNOWLEDGEMENTS

We are grateful to Wenbin Yu, Yunjuan Zuo, and Yunhong Tan from Xishuangbanna Tropical Botanical Garden, Chinese Academy of Sciences for their critical comments and help in genomic data analysis. We also thank Zhifeng Xu and Xiaoyun Wang from the Orchid Conservation Center of Yunnan Fengchunfang Biotechnology Co., Ltd, for kindly providing sample materials.

### Funding

This study was supported by the National Natural Science Foundation of China (Grant No. 31870183 and 32171655) and the West Light Talent Program of the Chinese Academy of Sciences (Grant No. E1XB011B01). The funders had no role in study design, data collection and analysis, decision to publish, or preparation of the manuscript.

### Grant Disclosures

The following grant information was disclosed by the authors:
The National Natural Science Foundation of China: 31870183, 32171655.
West Light Talent Program of the Chinese Academy of Sciences: E1XB011B01.

### Competing Interests

The authors declare there are no competing interests.

### Author Contributions

- Mengkai Li performed the experiments, analyzed the data, prepared figures and/or tables, authored or reviewed drafts of the article, and approved the final draft.
- Lu Tang performed the experiments, analyzed the data, prepared figures and/or tables, and approved the final draft.
- Jianping Deng performed the experiments, prepared figures and/or tables, and approved the final draft.
- Hanqing Tang analyzed the data, prepared figures and/or tables, and approved the final draft.

- Shicheng Shao analyzed the data, prepared figures and/or tables, and approved the final draft.
- Zhen Xing conceived and designed the experiments, authored or reviewed drafts of the article, and approved the final draft.
- Yan Luo conceived and designed the experiments, authored or reviewed drafts of the article, and approved the final draft.

## DNA Deposition

The following information was supplied regarding the deposition of DNA sequences:

The new sequences of cp genomes, nrDNA and chloroplast DNA sequences are available at GenBank: OM459712, OM524485, OM441937, OM441938, OM459823, OM459824.

## Data Availability

The sequence data are available in the Supplemental Files.

## New Species Registration

The following information was supplied regarding the registration of a newly described species:

Bulbophyllum pilopetalum M. K. Li, J. P. Deng & Y. Luo sp. nov.

urn:lsid:ipni.org:names:77311054-1

## Supplemental Information

Supplemental information for this article can be found online at http://dx.doi.org/10.7717/peerj.14721#supplemental-information.

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
