# Peer review of "Comparative chloroplast genomics of three species of Bulbophyllum section Cirrhopetalum (Orchidaceae), with an emphasis on the description of a new species from Eastern Himalaya"

_PeerJ, doi:10.7717/peerj.14721_

## Round 0.1 · original submission · Major Revisions

Please find the two reviews below and one of the reviewers included as well comments in the attached file. I agree with the reviewers in that you need to consider the latest classification by Hu et al and explain on the sections and on the controversial classification of Bulbophyllum. There are many problems with scientific names, please review them carefully and corroborate that they are in italics. In addition, I suggest the use of a company or a scientific English editor so that your paper reaches a suitable level.

·

Basic reporting

Dear Authors
Thanks for the submission of your manuscript.
Basic suggestions are:
In Abstract: In the section methods- Please check the spellings of Bulbophyllum and genus should be italic.
In the results section- Please add author citation with used species for comparison.

Experimental design

Methodology is good and just have to improve the grammar

Validity of the findings

The results revealed that the describes species in new to the science.

Additional comments

what about Preliminary conservation status?
Additional specimens examined (paratypes) ?

Reviewer 2 ·

Basic reporting

The manuscript has been written with unambiguous English throughout. A few specific comments and collections are made (see the attached PDF). Just to highly one serious mistake, the spelling of “Cirropetalum” (should be “Cirrhopetalum”) was wrong throughout the manuscript (including the title)!
The authors provided detailed literature review of the classification of Bulbophyllum and the Cirrhopetalum alliance, but not the section Cirrhopetalum. The authors should have indicated which concept of “section Cirrhopetalum” they adopted. As the authors cited Hu et al. (2020) have revealed that the majority of traditional sectional classification are highly artificial (Lines 102-104), which including the section Cirrhopetalum. But disappointingly, the authors did not refer to the latest classification, instead following the outdated reference, e.g., Flora of China (Chen & Vermulen, 2009). I would suggest the authors revisit and digest the latest reference, Hu et al. (2020) to improve the literature (Line 93-112) and indicate which concept of Cirrhopetalum the manuscript accepts and follows to assign the potential new species.

In general, the structure of the manuscript followed the journal’s standard. Figures and tables are relevant, while description of Figure 6 and Table 3 could be improved. (e.g., the title of Figure 6 could follow the format of Figure 7; Table 3: list the name of the three Bulbophyllum species).

Experimental design

New data was generated in this study with methods described with sufficient detail and information to replicate. However, the title (or the scope) looks too BIG compared to the dataset and comparative analyses conducted here. To my understanding, the two chloroplast genome data were generated to support the description of the potential new species, instead of a comparative study of the whole “section Cirrhopetalum”. It’s misleading to the reader. So, I would suggest the authors to re-write the title to match the scope of the study.

Validity of the findings

I think the underlying data have been provided and the analyses are statistically sound, for the description of the potential new species and the characteristics of chloroplast genome in three Bulbophyllum species. However, I don’t think these data and analyses could support the conclusions (Lines 66-70), which suddenly jumped into the topics of biogeography. The conclusions of biogeography pattern of the potential new species and allied species was not based on any analyses in the present study at all. Again, it’s out of the scope of this work and not linked to the original research questions, although the authors attempted to expand their discussion.

Additional comments

I would suggest the authors clearly identify the realistic research questions based on the available data and analyses; carefully revise the title, abstract, introduction and discussion. I believe major revision is necessary if the authors still target this journal or they may consider just publish the description of a new species on other specific journals focusing on new species description.

Annotated reviews are not available for download in order to protect the identity of reviewers who chose to remain anonymous.

---

## Round 0.2 · Minor Revisions

I appreciate your effort taking into account suggestions by reviewers. I suggest minor changes, please review all morphological terms such like type of inflorescence using a good dictionary in botany. Also review the order of structures in the description, and use the same format for separating plant organs, either using commas o semi-colons, but in a standard format. Give your manuscript a good review again, there are some singulars and plurals that are not well used.

---

## Round 0.3 · accepted · Accept

Thanks for the effort of changing the botanical terms and the rest of my suggestions. I am accepting the manuscript, however, I found a few misspellings of "inflorescence". When you review your proofs please check that in all cases is correct. I found that in several places you wrote "subumbellte" and this should be "sub-umbellate".